# The Effect of Hot Application Applied to the Breast with the Help of the Thera Pearl in the Postpartum Period on Mothers’ Milk Perception and Postpartum Breastfeeding Self-Efficacy: A Randomized Controlled Study

**DOI:** 10.3390/healthcare12100968

**Published:** 2024-05-08

**Authors:** Hava Ozkan, Betül Uzun Ozer, Ozlem Arı

**Affiliations:** 1Department of Midwifery, Faculty of Health Science, Ataturk University, 25240 Erzurum, Türkiye; havaoran@atauni.edu.tr; 2Department of Midwifery, Faculty of Health Science, Amasya University, 05000 Amasya, Türkiye; slymnkyr.25@gmail.com

**Keywords:** breast milk, breastfeeding, Thera Pearl, self-efficacy, perception of insufficient milk

## Abstract

Breastfeeding difficulties are often present in the first weeks of the postpartum period, and various factors can cause a mother’s early cessation of breastfeeding. Such factors include mothers’ self-efficacy and perception of insufficient milk. This study aimed to examine the effect of a hot application on the breast with the help of the Thera Pearl in the postpartum period on milk perception and breastfeeding self-efficacy. This was a randomized controlled trial. This study was completed with 64 puerperal mothers, 31 of whom were control and 33 were experimental. A Personal Information Form, Breastfeeding Observation Form, Breastfeeding Self-Efficacy Scale, and Insufficient Milk Perception Scale were used to collect data. The average breastfeeding self-efficacy scale scores of the experimental group were 42.46 and 62, respectively, at the first and last follow-up, while the control group’s mean scores were 57.42 and 47. The average scores of the experimental group on the insufficient milk perception scale were 19, 28, and 48, respectively, from the first to the last follow-up, while those of the control group were 28, 24, and 34, respectively. As a result, self-efficacy and milk perception are important factors affecting breastfeeding. In this study, we found that Thera Pearl application increased breastfeeding self-efficacy but did not affect milk perception.

## 1. Introduction

Breast milk is a nutrient with unique immunological and anti-inflammatory properties that protect against many diseases [1]. The World Health Organization (WHO) recommends starting breastfeeding immediately after birth and exclusively breastfeeding for the first 6 months. In addition, the WHO stated in its latest report that breastfeeding can continue after the age of 2 [2].

According to the United Nations Children’s Fund State of the World’s Children 2023 report, the rate of early breastfeeding initiation is 47%, and the rate of exclusive breastfeeding in the first six months is 48% [3]. According to the Turkish Demographic and Health Survey 2018 report, the rate of breastfeeding within the first hour after birth in Turkey is 71%, and the rate of exclusive breastfeeding for children under 6 months is 41% [4]. While the rate of early initiation of breastfeeding is high, breastfeeding is interrupted due to various problems, and the rate of exclusive breastfeeding decreases in the first six months. Studies have shown that the success of exclusive breastfeeding may depend on a woman’s confidence in her ability to perform this activity and the knowledge she acquires in the social environment [5,6]. To prevent breastfeeding from being interrupted before the first six months, it is necessary to know the reasons why mothers stop breastfeeding. In this case, it is important to help mothers and prevent early weaning. For example, a safe environment and support can prevent early weaning [7]. Therefore, various interventions to help sustain breastfeeding must be planned and implemented.

Breastfeeding difficulties are often present in the first weeks of the postpartum period, and various factors (age, education level, economic status, employment status, marital harmony, acceptance of the baby, etc.) can cause a mother’s early cessation of breastfeeding [8]. The mother’s self-efficacy and perception of insufficient milk are among such factors [9]. It is estimated that in many countries, mothers are unable to initiate or successfully continue breastfeeding because they do not believe they are producing enough milk [10]. Breastfeeding self-efficacy perception is the competence the mother feels regarding breastfeeding. It is known that mothers with high breastfeeding self-efficacy have fewer problems initiating and maintaining breastfeeding. In contrast, mothers with low breastfeeding self-efficacy cannot continue breastfeeding successfully, and breastfeeding is stopped at an early stage [11,12]. In addition, low breastfeeding self-efficacy may cause milk to be perceived as inadequate [9]. Studies on the subject show that a mother’s perception of milk is associated with breastfeeding self-efficacy [9,10].

In the absence or delayed onset of milk production after delivery, the mother may develop a perception of inadequate milk [13]. In this case, milk production is supported by using non-pharmacological methods [14]. Hot water application is often preferred to increase milk production [15]. However, when the relevant literature was scanned, no study was found to have used the Thera Pearl, and instead, heat application to the breast was generally performed through a warm towel or hot water bottle.

Although there are studies on breastfeeding self-efficacy and milk perception, no study has examined the effect of hot application applied to the breast with the help of the Thera Pearl on mothers’ milk perception and postpartum breastfeeding self-efficacy.

This study aimed to examine the effect of hot application on the breast with the help of the Thera Pearl in the postpartum period on milk perception and breastfeeding self-efficacy.

Research hypotheses:

**Hypothesis** **1** **(H1):**
*Thera Pearl’s application improves the perception of milk adequacy.*


**Hypothesis** **2** **(H2):**
*Thera Pearl’s application increases mothers’ breastfeeding self-efficacy.*


## 2. Materials and Methods

### 2.1. Design and Sample

This study was conducted as a randomized controlled trial. This research was conducted in the obstetric service of a university hospital in Turkey between October 2022 and May 2023. The research population consisted of women who gave birth for the first time in the gynecology service of this hospital between these dates. This study was conducted with 2 different groups (experimental and control). In order to calculate the sample size of the study, similar studies in the literature [16,17] were utilized. As a result of the power analysis performed according to the margin of error of 0.05, beta value of 0.02, and 95% confidence interval, it was determined that a minimum of 15 and a maximum of 35 puerperas should be included in each group. Considering the possibility of data loss from the groups, the sample size was increased by 25%, and data were collected from 88 individuals.

The total number of people evaluated for suitability for the study was 210. A total of 86 individuals did not meet the inclusion criteria, and 36 refused to participate. Randomization was performed with 88 people, and each group consisted of 44 people. The groups were informed about the research, and the experimental group was given intervention material. During data collection, 11 people from the experimental group were excluded (5 people who stopped breastfeeding completely and 6 people who did not want to apply the Thera Pearl). In the control group, 13 people were excluded (4 people with missing contact information, 3 people who stopped breastfeeding completely, and 6 people who did not want to continue the research). As a result, this study was completed with 64 mothers (33 experimental and 31 control). In addition, after the completion of the data-collection phase, post hoc power analysis was performed with the G-Power Data Analysis program (GPower 3.1). When the power analysis was calculated by taking a 95% confidence interval, *p* = 0.05 significance level, and sample size of 64, the power of the study was determined to be 95%, and it was concluded that the sample had good power to represent the population.

This study included postpartum mothers who were primary school graduates between the ages of 18–45, primiparous, and agreed to participate in the research. Postpartum women who did not meet these criteria were excluded from the study. Mothers who met the criteria were invited to participate in the study and were provided with a comprehensive information form detailing the potential risks and benefits of the trial and the procedures and interventions involved. Those who agreed to participate then signed the informed consent. While the independent variable of the research is Thera Pearl application, the dependent variable is mothers’ breastfeeding self-efficacy levels and perception of milk sufficiency. Outcome variables were evaluated separately by the primary investigator. The second and third researchers conducted the data-collection phase (baseline, end of trial, and follow-up) together. The research team was not blinded.

### 2.2. Data Collection

A Personal Information Form, Breastfeeding Observation Form, Breastfeeding Self-Efficacy Scale (BSS), and Insufficient Milk Perception Scale (IMPS) were used to collect data (Appendix A).

Personal Information Form: This form consists of 13 questions created by the researcher by scanning the literature and includes the sociodemographic characteristics of the participants.

Breastfeeding Information Form: The researchers created this form by scanning the literature. It contains various questions regarding breastfeeding.

BSS: This is a scale developed by Dennis and Faux to evaluate the breastfeeding self-efficacy levels of mothers, the first form of which has 33 items [18]. Later, a 14-item short-form scale was developed [19].

IMPS: This scale, developed by McCarter-Spaulding in 2001 to determine the perception of insufficient breast milk, consists of 6 questions. The scale can be scored between 0 and 50 points. A high total score indicates that the perception of milk adequacy increases [20].

### 2.3. Study Procedure

Postpartum women included in this study were randomized using single-blind randomization. Red (experimental) and white (control) colored balls were placed in a closed box. There were 88 balls in the box, 44 red and 44 white. All postpartum women were asked to randomly choose a ball from the box individually, and the selected ball was placed back in the box. If the selected ball was red, the individual was assigned to the postpartum experiment group, and if the ball was white, the individual was assigned to the control group. This cycle continued until the desired number was reached for the experimental and control groups.

#### 2.3.1. Experimental Group

The first interview with women who gave birth was held in the postpartum service of the relevant hospital. During this meeting, the Personal Information Form, Breastfeeding Observation Form, BSS, and IMPS were filled out. At the same time, during this meeting, the telephone and address information of the participants were obtained on postpartum days 2–7 and 10–15. The mothers were informed that a home visit would be made in the following days. The Thera Pearl application was explained to the mothers in detail, and the first application was performed together in the hospital.

Mothers who continued were visited once a day, 2–7 days, and 10–15 days after birth. During these follow-ups, their application of the Thera Pearl was examined. The BSS and the IMPS were completed again.

Intervention material: Thera Pearl (Figure 1).

Information about the application: Thera Pearl thermogel packs for hot therapy can be heated in the microwave for a maximum of 15 s inside their fabric covers or in hot water by removing their fabric covers. The packages should be kept in boiled water for 1–2 min after removing their covers and then applied by placing them in the case.

This application should be performed 4 times per day: in the morning, at noon, in the evening, and at night before going to bed. Each application should be applied to both breasts and last at least 20 min.

#### 2.3.2. Control Group

No intervention was applied to the control group. The first interview with women who gave birth was held in the postpartum service of the relevant hospital. During this meeting, the Personal Information Form, Breastfeeding Observation Form, BSS, and IMPS were filled out. During the first follow-up, the mothers’ phone numbers and address information were obtained. It was said that they would be visited once at 2–7 days and once at 10–15 days after birth.

Mothers who continued were visited once per day, 2–7 days, and 10–15 days after birth. The BSS and the IMPS were completed again.

Follow-up of mothers in the control group continued until the 15th postpartum day. In certain situations encountered during the research period, mothers were excluded from the study. For example, the biggest problem encountered during data collection was that mothers completely stopped breastfeeding and started using formula. Three people in these circumstances were excluded from the control group. Another problem encountered is that mothers who were affected by the questions in the data collection forms wanted to receive breastfeeding support from researchers regarding breastfeeding. However, due to the study principle, the researchers did not provide any support for breastfeeding in either group. Only hot water packs were given to the experimental group. For this reason, no support was provided to the control group. In this case, some mothers stated that they did not want to continue the study and withdrew (6 mothers).

A flow diagram of the experiment is shown in Figure 2.

### 2.4. Ethical Considerations

This study followed the Consolidated Standards of Reporting Trials (CONSORT) reporting guideline [21] and was performed in line with the principles of the Declaration of Helsinki. This study was approved by the university’s Clinical Research Ethics Board (IRB-B.30.2.ATA.0.01.00/410). The women were informed about the study, and their verbal and written consent was obtained. All women were told that all collected data would be kept confidential.

This research can be found in a study protocol registration system from the United States. It has been registered on the “ClinicalTrials.gov” website, an important database for the registration of pharmaceutical and interventional studies, with ID: NCT05806892.

### 2.5. Statistical Analysis

Data were analyzed in IBM SPSS Statistics for Windows, version 23.0 (IBM SPSS Corp., Armonk, NY, USA). The suitability of the data for normal distribution was examined with Shapiro–Wilk tests. The Mann–Whitney U test was used to compare data that did not show a normal distribution in pairs. The Friedman test was used to compare data that did not comply with a normal distribution three or more times, and multiple comparisons were examined with the Dunn test. The Pearson correlation coefficient was used to examine the relationship between scale scores that conform to a normal distribution. Spearman’s rho correlation coefficient was used to examine the relationship between scale scores that did not comply with a normal distribution. Analysis results were presented as a frequency (percentage) for categorical variables, mean ± standard deviation, and median (minimum–maximum) for quantitative variables. The effect of independent variables on the changes in scores was examined by linear regression analysis. The significance level was taken as *p* < 0.050.

## 3. Results

Table 1 shows the comparison of demographic characteristics by groups. In this study, no statistically significant difference was obtained between the experimental and control groups and the distributions of sociodemographic characteristics (*p* > 0.050). A total of 41.9% of the control group was between the ages of 26 and 30, and 42.4% of the experimental group was between the ages of 20 and 25. In addition, it was determined that most of the participants in both groups lived in the province, were primary and secondary school graduates, did not work, had a nuclear family structure, and had income equal to their expenses.

Table 2 shows the distribution of the answers given by postpartum mothers to breastfeeding observation definition questions. In this study, no statistically significant difference was obtained between the experimental and control groups and the distributions of the parameters (*p* > 0.050). When breast fullness was evaluated at the beginning of the breastfeeding process, no statistically significant difference was seen between the experimental and control groups (*p* = 0.796). Another important finding in Table 2 is that while breastfeeding was successful in the first half hour in both the experimental and control groups, more than half of the babies in both groups were given formula.

For the control group, no statistically significant difference was obtained in the BSS median score during follow-ups at different times (*p* = 459). However, a statistically significant difference was obtained between the BSS median scores of the experimental group (*p* < 0.001). In the experimental group, the first follow-up BSS median score was 42, the second follow-up median value was 46, and the third follow-up median score was 62. According to the groups, while there was no statistically significant difference between the BSS median score of the first and second follow-up (*p* > 0.050), a statistically significant difference was obtained between the BSS median score of the third follow-up (*p* = 0.013). While the median BSS score at the third follow-up was 47 in the control group, this value was 62 in the experimental group (Table 3).

Table 4 shows no statistically significant difference between the IMPS median score of the first, second, and third follow-up according to the experimental and control groups (*p* > 0.050). However, there was a statistically significant difference between the IMPS median score of the experimental group according to time (*p* < 0.001). It was determined that the IMPS median score increased from the first follow-up to the third follow-up in the experimental group. There was no statistically significant difference between the IMPS median score of the control group according to time (*p* = 0.516).

The linear regression model established to examine the effect of independent variables on the change score of BSS second follow-up (F = 2.691; 0.007) and third follow-up (F = 3.424; 0.001) was found to be statistically significant. At the second follow-up, the independent variables explain 24% of the variance in the BSS score, and at the third follow-up, 32%. Educational status in the second follow-up, and education and income status in the third follow-up, affected the follow-up change score (Table 5).

The linear regression model established to examine the effect of independent variables on the IMPS second follow-up (F = 2.482; 0.012) and third follow-up (F = 2.875; 0.004) change score was found to be statistically significant. At the second follow-up, the independent variables explained 22% of the variance in the IMPS score, and at the third follow-up, 26%. In the second and third follow-up, education and income status affected the follow-up change score (Table 5).

## 4. Discussion

Initiating breastfeeding within the first 24 h after birth is the gold standard in infant nutrition [22]. Therefore, breastfeeding should be initiated and supported in the first 24 h postpartum. In particular, identifying and correcting mothers’ faulty breastfeeding behaviors has an important place in the initiation and continuation of breastfeeding [23]. In this study, we saw that most of the participants in the experimental and control groups started breastfeeding within the first half hour and used formula despite milk secretion. Participants stated that their milk production was insufficient, so they used formula. However, when we examined the fullness of the breasts, milk secretion, and the baby’s latching status in both groups in the first 24 h, we found that it was unnecessary to start formula use (Table 2). This made us think that the mothers who participated in our study had a perception of insufficient milk, so they started formula feeding earlier. In another study similar to ours, it was found that although only 5% of women had insufficient milk physiologically, 50% had the perception of insufficient milk [24]. These results are thought to be an important determinant of breastfeeding self-efficacy. If a mother’s breastfeeding self-efficacy level is low, it may cause her to perceive her milk supply as insufficient, supplement her insufficient breast milk with formula, decrease her actual milk production, and eventually stop breastfeeding [20].

Hot application to the breast during breastfeeding is preferred only to increase lactation and eliminate breast problems (engorgement, clogged milk ducts, mastitis, breast abscess, infection, nipple sensitivity/pain, cracked nipple, etc.) [25,26]. However, in addition to physiologically increasing lactation and eliminating breast problems, these practices also have some psychological effects. It is thought that they accelerate the readiness of women in particular. The amount of milk produced by a mother who is psychologically ready for the breastfeeding process and whose self-efficacy increases will also increase [27]. Unfortunately, no study has been found in the literature showing the long-term psychological effects of hot application. In this study, breastfeeding self-efficacy increased from the first to the third follow-up in the experimental group in which the Thera Pearl was applied. However, this was not the case for the control group. In addition, at the last follow-up of the study, the breastfeeding self-efficacy of the experimental group was found to be significantly higher than the control group. Since hot application to the breast increases milk production, the self-efficacy of women in the experimental group may have increased. In addition, mothers’ comfort levels may have been increased with warm application, thus improving their self-efficacy perception. As a matter of fact, it has been accepted that there is a connection between mothers’ comfort levels and their perception of self-efficacy, where mothers with high comfort levels have higher breastfeeding success [28]. Hot application to the breast especially falls within the sociocultural comfort dimension. Sociocultural comfort is based on providing care and counseling in accordance with the person’s traditions and providing home care [29,30]. In our country, applying heat to the breast during breastfeeding has become a tradition. In addition, monitoring mothers and providing home care during the first 15 days, which is one of the most important postpartum periods, may have affected the study results.

In this randomized controlled trial, we saw that the perception of insufficient milk did not change depending on the intervention between the groups. Still, the perception of milk improved from the first to the third follow-up in the experimental group where the Thera Pearl was applied. This result may be because hot application increases breast milk. On the other hand, the lack of difference between the groups is thought to be due to the women’s lack of knowledge about breast milk and breastfeeding. As a matter of fact, it is known that breastfeeding education has a positive effect on the perception of insufficient milk [31].

Additionally, this study showed that some sociodemographic characteristics (education and income status) have an impact on breastfeeding self-efficacy and the perception of insufficient milk. In this regard, it is important to conduct studies with adolescents and immigrants, especially those who are disadvantaged in society, in terms of the debatability of this finding.

This trial has several limitations. First, the small sample should be increased in future studies to validate these findings. In this study, short-term follow-up was performed. Therefore, it is recommended to conduct studies with longer follow-up periods.

## 5. Conclusions

This study has shown that self-efficacy and milk perception affect breastfeeding. In this study, we found that the Thera Pearl’s application increased breastfeeding self-efficacy. Considering that mothers with high self-efficacy can cope better with the difficulties they face, think positively, and prefer breastfeeding, our study reached its goal.

The perception of insufficient milk is a major obstacle for mothers trying to exclusively breastfeed for the first 6 months. It is said that the perception of insufficient milk plays an important role in stopping breastfeeding in many societies worldwide. In this study, we found that milk perception did not change in the experimental group due to Thera Pearl’s application. This result is likely due to the lack of information about breastfeeding. In this regard, we recommend conducting studies in larger sample groups where Thera Pearl application and breastfeeding education are given together. In addition, we recommend conducting qualitative research examining women’s breastfeeding experience during the postpartum period, breastfeeding attitudes, and affecting factors.

These findings show that hot application to the breast in the postpartum period increases breastfeeding self-efficacy but does not affect milk perception. The perception of insufficient milk is one of the most important reasons why breastfeeding is interrupted. Therefore, the result we obtained from our study proved that intervention alone is not effective in improving mothers’ milk perception.

## Figures and Tables

**Figure 1 healthcare-12-00968-f001:**
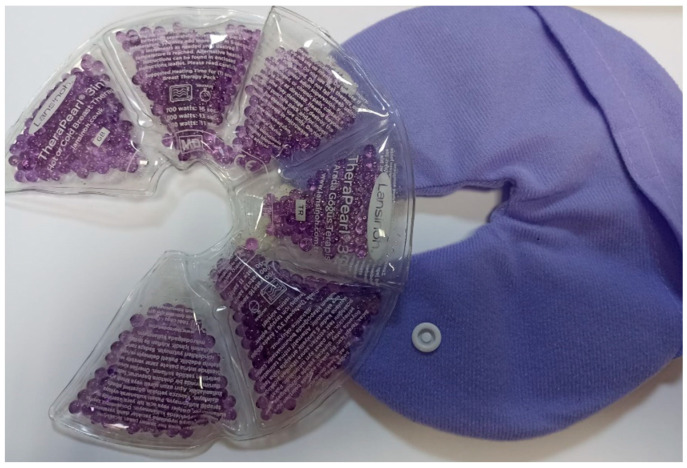
Thera Pearl.

**Figure 2 healthcare-12-00968-f002:**
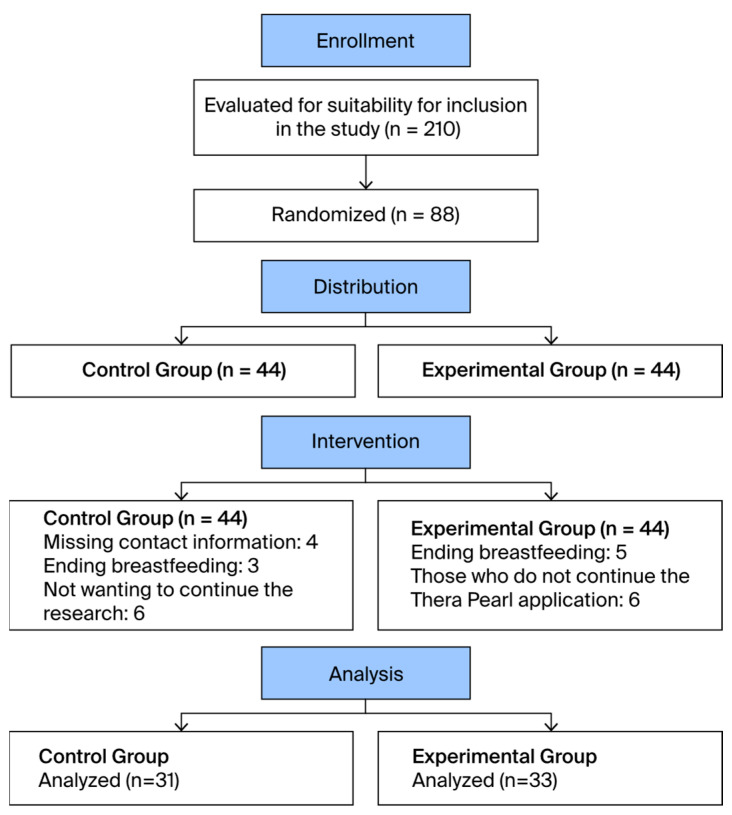
Flow diagram of the experimental procedure.

**Table 1 healthcare-12-00968-t001:** Comparison of demographic characteristics by group (n = 64).

	Group			Test Statistics	*p*-Value
	Control (n = 31) Mean (SD)/N (%)	Experiment (n = 33) Mean (SD)/N (%)	Total		
Age					
20–25	9 (29)	14 (42.4)	23 (35.9)	1.694	0.638
26–30	13 (41.9)	10 (30.3)	23 (35.9)		
31–35	6 (19.4)	7 (21.2)	13 (20.3)		
36 and over	3 (9.7)	2 (6.1)	5 (7.8)		
Living Place					
City	21 (67.7)	17 (51.5)	38 (59.4)	1.760	0.415
County	6 (19.4)	10 (30.3)	16 (25)		
Village	4 (12.9)	6 (18.2)	10 (15.6)		
Educational Status					
Primary school	8 (25.8)	9 (27.3)	17 (26.6)	1.614	0.656
Middle school	11 (35.5)	8 (24.2)	19 (29.7)		
High school	5 (16.1)	9 (27.3)	14 (21.9)		
University	7 (22.6)	7 (21.2)	14 (21.9)		
Working Status					
I am working	8 (25.8)	6 (18.2)	14 (21.9)	0.189	0.664
I am not working	23 (74.2)	27 (81.8)	50 (78.1)		
Family structure					
Nuclear family	23 (74.2)	29 (87.9)	52 (81.3)	1.169	0.280
Extended family	8 (25.8)	4 (12.1)	12 (18.8)		
Economical situation					
Income Less Than Expenses	4 (12.9)	9 (27.3)	13 (20.3)	2.369	0.306
Income Equals Expenditures	24 (77.4)	20 (60.6)	44 (68.8)		
Income More Than Expenditures	3 (9.7)	4 (12.1)	7 (10.9)		

**Table 2 healthcare-12-00968-t002:** Distribution of answers of postpartum women to breastfeeding observation and identification questions (n = 64).

	Group			Test Statistics	*p*-Value
	Control (n = 31) Mean (SD)/N (%)	Experiment (n = 33) Mean (SD)/N (%)	Total		
Are the breasts full?					
Yes	17 (54.8)	16 (48.5)	33 (51.6)	0.067	0.796
No	14 (45.2)	17 (51.5)	31 (48.4)		
Was the baby breastfed in the first half hour?					
Breastfed successfully	16 (51.6)	15 (45.5)	31 (48.4)	0.444	0.801
Breastfed with assistance	8 (25.8)	11 (33.3)	19 (29.7)		
Breastfeeding could not be achieved	7 (22.6)	7 (21.2)	14 (21.9)		
Has the baby been given formula?					
Yes	18 (58.1)	23 (69.7)	41 (64.1)	0.502	0.479
No	13 (41.9)	10 (30.3)	23 (35.9)		
If formula is given, what is the reason?					
Lack of milk secretion	16 (88.9)	15 (65.2)	31 (75.6)	6.316	0.177
Hypoglycemia	1 (5.6)	4 (17.4)	5 (12.2)		
Hyperbilirubinemia	0 (0)	3 (13)	3 (7.3)		
Baby’s inability to latch onto the breast	1 (5.6)	0 (0)	1 (2.4)		
Sunken nipple	0 (0)	1 (4.3)	1 (2.4)		
How does the baby latch on?					
Good	17 (54.8)	19 (57.6)	36 (56.3)	0.192	0.908
Middle	8 (25.8)	7 (21.2)	15 (23.4)		
Weak	6 (19.4)	7 (21.2)	13 (20.3)		
Has milk secretion started?					
Comes after milking by hand	6 (19.4)	9 (27.3)	15 (23.4)	0.597	0.742
Baby breastfeeds easily	16 (51.6)	16 (48.5)	32 (50)		
No milk release	9 (29)	8 (24.2)	17 (26.6)		

**Table 3 healthcare-12-00968-t003:** Comparison of BSS total scores of experimental and control groups according to observations (n = 64).

	Group				Test Statistics	*p*-Value *
	Control		Experiment			
	Mean ± S.D.	Mean (Min–Max)	Mean ± S.D.	Mean (Min–Max)		
After birth (in the first 24 h)	47.29 ± 18.18	57 (14–68)	44.12 ± 18.79	42 (14–70)	462	0.506
2–7 days after birth	43.58 ± 15.47	42 (14–75)	46.64 ± 15.27	46 (14–70)	443.5	0.361
10–15 days after birth	46.94 ± 19.05	47 (14–70)	59.3 ± 12.84	62 (14–70)	328.5	0.013
Test statistics	1.556		24.555			
*p* **	0.459		<0.001			

* Mann–Whitney U test; ** Friedman test.

**Table 4 healthcare-12-00968-t004:** Comparison of IMPS total scores by groups and times (n = 64).

	Group				Test Statistics	*p*-Value *
	Control		Experiment			
	Mean ± S.D.	Mean (Min–Max)	Mean ± S.D.	Mean (Min–Max)		
After birth (in the first 24 h)	26.87 ± 15.99	28 (0–50)	21.85 ± 14.63	19 (0–47)	415	0.195
2–7 days after birth	26.97 ± 14.52	24 (1–50)	29.7 ± 14.35	28 (3–50)	444	0.364
10–15 days after birth	30.06 ± 17.96	34 (0–50)	39.45 ± 12.22	43 (7–50)	384	0.083
Test statistics	1.322		34.063			
*p* **	0.516		<0.001 *			

* Mann–Whitney U test; ** Friedman test.

**Table 5 healthcare-12-00968-t005:** Examining the effect of independent variables with linear regression analysis (n = 64).

		β^1^ (%95 CI)	S. D.	β^2^	Test Statistics	*p*	VIF	F; *p*	R^2^	Adjusted R^2^
BSS (2–7 days after birth)	Stationary	−13.008 (−28.529–2.514)	7.731		−1.682	0.099		2.691; 0.007	0.388	0.244
	Group (Reference: Control)	−6.411 (−14.22–1.397)	3.889	−0.197	−1.648	0.105	1.188			
	Age (Reference: 20–25) 26–30	−1.502 (−10.393–7.389)	4.429	−0.044	−0.339	0.736	1.420			
	31 and over	8.234 (−2.051–18.519)	5.123	0.227	1.607	0.114	1.668			
	Educational Status (Reference: Primary School)									
	Middle school	8.523 (−1.668–18.713)	5.076	0.239	1.679	0.099	1.691			
	High school	11.885 (0.662–23.107)	5.590	0.302	2.126	0.038	1.679			
	University	21.115 (8.886–33.344)	6.091	0.536	3.466	0.001	1.994			
	Income status (Income less than expenses)									
	Income equals expenses	−2.458 (−12.368–7.452)	4.936	−0.070	−0.498	0.621	1.646			
	Income exceeds expenses	−14.518 (−30.97–1.935)	8.195	−0.278	−1.771	0.082	2.057			
BSS (10–15 days after birth)	Stationary	−9.238 (−26.338–7.862)	8.518		−1.085	0.283		3.424; 0.001	0.446	0.316
	Group (Reference: Control)	−17.561 (−26.163–8.958)	4.285	−0.465	−4.098	<0.001	1.188			
	Age (Reference: 20–25) 26–30	−6.209 (−16.004–3.586)	4.879	−0.158	−1.273	0.209	1.420			
	31 and over	5.408 (−5.923–16.739)	5.644	0.129	0.958	0.342	1.668			
	Educational Status (Reference: Primary School)									
	Middle school	7.7 (−3.527–18.927)	5.592	0.187	1.377	0.175	1.691			
	High school	11.551 (−0.812–23.915)	6.158	0.253	1.876	0.066	1.679			
	University	23.703 (10.23–37.175)	6.711	0.520	3.532	0.001	1.994			
	Income status (Income less than expenses)									
	Income equals expenses	−6.104 (−17.022–4.814)	5.438	−0.150	−1.122	0.267	1.646			
	Income exceeds expenses	−19.511 (−37.637–1.385)	9.029	−0.323	−2.161	0.035	2.057			
IMPS (2–7 days after birth)	Stationary	−10.074 (−23.146–2.998)	6.511		−1.547	0.128		2.482; 0.012	0.369	0.220
	Group (Reference: Control)	−7.989 (−14.565–1.413)	3.276	−0.296	−2.439	0.018	1.188			
	Age (Reference: 20–25) 26–30	−3.607 (−11.095–3.881)	3.730	−0.128	−0.967	0.338	1.420			
	31 and over	5.118 (−3.544–13.78)	4.315	0.170	1.186	0.241	1.668			
	Educational Status (Reference: Primary School)									
	Middle school	8.506 (−0.076–17.089)	4.275	0.288	1.990	0.052	1.691			
	High school	11.34 (1.889–20.791)	4.708	0.347	2.409	0.020	1.679			
	University	16.456 (6.157–26.755)	5.130	0.504	3.208	0.002	1.994			
	Income status (Income less than expenses)									
	Income equals expenses	−3.534 (−11.88–4.812)	4.157	−0.121	−0.850	0.399	1.646			
	Income exceeds expenses	−14.43 (−28.286–−0.574)	6.902	−0.334	−2.091	0.042	2.057			
IMPS (10–15 days after birth)	Stationary	−12.083 (−27.847–3.681)	7.852		−1.539	0.130		2.875; 0.004	0.404	0.263
	Group (Reference: Control)	−15.174 (−23.104–7.243)	3.950	−0.453	−3.841	<0.001	1.188			
	Age (Reference: 20–25) 26–30	−3.967 (−12.997–5.063)	4.498	−0.114	−0.882	0.382	1.420			
	31 and over	8.806 (−1.64–19.251)	5.203	0.236	1.692	0.097	1.668			
	Educational Status (Reference: Primary School)									
	Middle school	7.188 (−3.162–17.538)	5.155	0.196	1.394	0.169	1.691			
	High school	6.808 (−4.59–18.206)	5.677	0.168	1.199	0.236	1.679			
	University	13.253 (0.833–25.673)	6.187	0.327	2.142	0.037	1.994			
	Income status (Income less than expenses)									
	Income equals expenses	−6.024 (−16.089–4.041)	5.013	−0.167	−1.202	0.235	1.646			
	Income exceeds expenses	−17.146 (−33.856–0.437)	8.323	−0.320	−2.060	0.045	2.057			

β^1^ (%95 CI): unstandardized beta coefficient (%95 CI), β^2^: standardized beta coefficient.

## Data Availability

The data presented in this study are available on request from the corresponding author.

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
