# Peer review of "The Effect of Hot Application Applied to the Breast with the Help of the Thera Pearl in the Postpartum Period on Mothers’ Milk Perception and Postpartum Breastfeeding Self-Efficacy: A Randomized Controlled Study"

_healthcare, 2024, doi:10.3390/healthcare12100968_

Round 1

Reviewer 1 Report

Comments and Suggestions for Authors

Thank you very much for conducting this important study. Interventions to increase breastfeeding knowledge, practice, self-efficacies, and norms are critical, especially when women and family members are bombarded with messages and free samples promoting commercial milk formula. I have some comments for your consideration to improve the study findings. The comments about data analysis and interpretation are critical ones.

1.         Lines 36-37: The authors might want to use more recent data from WHO/UNICEF or the global nutrition target.

2.         Lines 55-56: It would be better to use the term "association" rather than making a causal statement because perception may influence IYCF practices, which in turn may affect milk production.

3.         Line 68: The statement of H1 is not very clear.

4.         Line 72 and other text: Please use consistent terminology such as "randomized controlled trial" or "randomized case control."

5.         The description of sample size determination is not very clear. The authors need to estimate the sample size for each group based on the anticipated difference between the two groups. Please mention the assumptions and basis for these assumptions.

6.         Line 84: Do the authors mean "cannot be contacted"? "Bias" is typically used in a different context.

7.         Please include an appendix with all data collection forms.

8.         Regarding the study procedure, please specify how women were approached and invited to participate in the study. What was the acceptance rate? This stage is before obtaining the 88 women. Who were the ones who performed data collection and follow up visits?

9.         Please provide a description of study variables and indicators.

10.       Please include the statistical data analysis section. For this type of study, I have seen that people use difference-in-difference analysis. Also, use intention-to-treat analysis (use data from initial randomization intervention of 41 vs control of 41).

11.       Line 121 and other places: I think the authors mean "the mother/woman" instead of "he."

12.       Tables 1-4: Please limit "*" for statistical difference. Please use "." instead of "," to indicate decimal places.

13.       Please specify each of the abbreviations upon first use.

Author Response

First of all, we, are grateful to you for taking the time and evaluating our article. Your evaluations contributed greatly to my academic development. Thank you very much. I tried to follow all your suggestions carefully. I hope I was successful.

I received evaluations from 3 referees. I have shown the corrections you requested in YELLOW. Some fixes were common.

Thank you again for your interest and wish you a good day...

  1. Lines 36-37: The data presented in this section has been renewed by examining current reports.
  2. Lines 55-56: In line with the suggestion, the term "relationship" was used and the relevant sentence was edited accordingly.
  3. Line 68: Hypotheses are written descriptively.
  4. Line 72 and other text: "randomized controlled trial" was used.
  5. Sample size calculation explained.
  6. Line 84: Sometimes I could not contact the postpartum women because they gave the wrong phone number and address. I will explain this paragraph in detail in the flow chart.
  7. Data collection forms have been added.
  8. The procedure and data collection part of the research was detailed.
  9. The variables are shown.
  10. Statistical data analysis section added.
  11. Line 121: "he/she" corrected to "mother".
  12. In the tables, significant results were delimited with (*) and a period was used for decimals.
  13. Abbreviations corrected.

Reviewer 2 Report

Comments and Suggestions for Authors

Dear authors, the paper presents a well-designed study on the important issue of breastfeeding support. The rationale for conducting the study is sound. The study has bioethical approvals.

The strength of this manuscript is a topic that has not been studied in detail before (the use of Thera Pearl). The topic of breastfeeding abandonment due to an apparent lack of breastmilk is an important practical problem in the field of obstetrics. The study was correctly planned and constructed. The results are presented clearly, and conclusions are drawn from the results.

However, there remain minor shortcomings that need improvement:

Verse 75 - shouldn't it be obstetric and not gynecological?

Verse 125 -127 please check the correctness of these sentences and translations 

Verse 148-149, 281-290- please check the correctness of these sentences and translation

Verse 153 changes the tiltle of the fig.

The numbers on the flowchart do not match, please check their correctness

- add an appropriate checklist such as CONSORT. Please add what modifying and confounding variables were included.
- Please improve the flow chart (Fig. 2). It shows that out of 88 subjects, 4 dropped out, and thus 82 remained, yet 88-4=84. The same is true for the calculation of the other groups, or please state that the exclusion factors may have been multiple choice.

This sentence is not understandable " Then the participant is 2-7 postpartum. and 10-15. She was visited at home once a day, and the Breastfeeding Self-Efficacy Scale and the Insufficient Milk Perception Scale were filled out again."

please, give me the meaning of "Figure 2 Consort Diagram."

Author Response

First of all, we, are grateful to you for taking the time and evaluating our article. Your evaluations contributed greatly to my academic development. Thank you very much. I tried to follow all your suggestions carefully. I hope I was successful.

I received evaluations from 3 referees. I have shown the corrections you requested in RED. Some fixes were common.

Thank you again for your interest and wish you a good day...

  1. Verse 75: I corrected the word to "obstetrics clinic".
  2. Verse 125 -127: Sentence corrected. Dear Sir, I will ask the journal to make language edits to correct translation errors.
  3. Verse 148-149, 281-290: Corrected
  4. "Flow diagram of the experimental procedure" has been redone. All mistakes have been corrected.
  5. I have revised and edited all errors resulting from the translation.
  6. 2: I edited it as "Flow diagram of the experimental procedure".
  7. Consort list added.

Reviewer 3 Report

Comments and Suggestions for Authors

The paper titled “The Effect of Hot Application Applied to the Breast with the Help of Thera Pearl in the Postpartum Period on Mothers' Milk Perception and Postpartum Breastfeeding Self-Efficacy: Randomized Controlled Study” shows the results of a clinical trial to examine the effect of applying heat to the maternal breast "Thera Pearl" in the postpartum period to verify the impact on the perception of milk and breastfeeding self-efficacy.

I believe that it is relevant to investigate maternal perceptions of the breastfeeding process and their self-efficacy. There are some comments that may help the authors to improve the presentation of their research and results.efficacy.

Author Response

First of all, we, are grateful to you for taking the time and evaluating our article. Your evaluations contributed greatly to my academic development. Thank you very much. I tried to follow all your suggestions carefully. I hope I was successful.

I received evaluations from 3 referees. I have shown the corrections you requested in GREEN. Some fixes were common.

Thank you again for your interest and wish you a good day...

  1. Since our research was supported by a project, it was not possible to change the title. But I will take your valuable suggestion into consideration in all my future work.
  2. Quantitative information added to the summary
  3. Line 45: error corrected
  4. Line 59: dear sir, thera pearl is the name given to the hot water packs currently sold in our country and used in the postpartum period. It is the product of a trademark.
  5. Recommended articles have been added to the appropriate sections. Thank you also for your current article suggestions.
  6. The mistake made in the sample calculation has been corrected.
  7. Line 125, 133, 146: Attempts were made to correct spelling mistakes and translation errors. Help will be taken from the journal for better translation.
  8. Line 148: some mothers had to be excluded from the control group while collecting the research data. For example, people with insufficient milk secretion started using formula and stopped breastfeeding completely. In this case, the mother was excluded from the study. Additionally, there were mothers who wanted to receive breastfeeding support from the researchers and when they could not receive support, they withdrew from the study. I added it to my article.
  9. Line 173 and 183: some adjustments have been made to the tables.
  10. Line 193 and Line 201: Dear Sir, the typo has been corrected and the abbreviations have been revised.
  11. A recommendation for qualitative studies was added to the conclusion section.
  12. Line 279, 286, 288: missing information completed.

Round 2

Reviewer 1 Report

Comments and Suggestions for Authors
  1. Citations:
    • Please ensure that the Breastfeeding Self-Efficacy Scale (BSE) and the Insufficient Milk Perception Scale (IMPS) cite the original articles where they were first introduced.
  2. Data Collection Tools:
    • The Breastfeeding Observation Form seems to be structured more like a questionnaire. Consider revising the naming of the form.
  3. Insufficient Milk Perception Scale:
    • The translation of the instructions for the scale needs verification. The current wording may be improved for clarity: “This scale includes questions about your perceptions of your milk supply. Respond ‘yes’ or ‘no’ to the initial question about your current milk supply. Rate subsequent questions on a scale from 0 to 10, where 0 signifies no milk and 10 signifies an excessive supply. Scores closer to 0 suggest perceived insufficient milk, while scores closer to 10 suggest perceived sufficiency.”
  4. Statistical Analysis:
    • The statistical analysis methods, particularly for Tables 3 and 4, are not clearly stated. A footnote in these tables should specify the statistical tests used.
    • The sample size for each follow-up period should be included for clarity.
  5. Consistency in Naming:
    • Ensure consistent naming of the follow-up periods in both the methods section and the tables. Consider adding a footnote to the tables for further clarification.
  6. Analysis Recommendations:
    • It is advisable to apply difference-in-difference analyses to allow for the control of baseline difference and socio-economic characteristics in certain regression models.
  7. Discussion of Limitations:
    • The last paragraph contains elements that would be more appropriate in the limitations section. The discussion on limitations should be expanded to provide a comprehensive overview.
  8. Implications of the Study:
    • After the conclusions paragraph, include a section discussing the implications of the study’s findings.

Author Response

Your valuable suggestions strengthened my research. That's why I'm grateful to you. I tried to make all the changes you requested. I hope I was successful. I colored my corrections in YELLOW.

Thank you.

  1. Citations: Sources regarding the scales have been corrected. All sources in the article have been checked.
  2. Data Collection Tools: "The Breastfeeding Observation Form" has been changed to "Breastfeeding Information Form".
  3. Insufficient Milk Perception Scale: IMPS scale instructions have been corrected.
  4. Statistical Analysis: Added statistical analysis methods for Tables 3 and 4. added sample size.
  5. Consistency in Naming: Follow-up periods were shown with a single expression in the methods and tables.
  1. Analysis Recommendations: The requested analysis has been made.
  1. Discussion of Limitations: Limitations are discussed.
  2. Implications of the Study: The Implications section has been added.

Reviewer 3 Report

Comments and Suggestions for Authors

After reviewing the changes and implementing the comments that had been indicated to the authors to improve the presentation of their manuscript, I consider that a good job has been done and the manuscript has improved considerably. However, I would like to add that the title still seems too long to me

Author Response

Your valuable suggestions strengthened my research. That's why I'm grateful to you.